# Dynamic Compression Strategies for Uniform Low-Dimensional Representations in Human Brain and Neural Network

## Abstract

Recent studies suggest that the generalization performance of neural networks is strongly linked to their ability to learn low-dimensional data representations. However, limited attention has been given to the consistency of compression across different types of input data. In this work, we compute the intrinsic dimensions of raw data and their corresponding representations to quantify the extent of information compression in neural networks. Our results indicate that the pre-trained model CLIP compresses complex datasets significantly more than simpler ones and tends to represent diverse datasets with uniform low-dimensional manifolds. Similarly, we observe stable dimensionality in neural manifolds in the brain across various tasks and cognitive processes, suggesting that biological systems also favor consistent low-dimensional representations. Theoretically, we demonstrate that lower-dimensional manifolds increase the probability of interpolation, facilitating the representation of new samples as convex combinations of existing data. Additionally, we derive an upper bound on generalization error within the interpolation regime, which tightens as the dimensionality of the data decreases. These findings underscore the critical role of uniform low-dimensional manifolds in supporting efficient and generalizable information representation in both artificial and biological neural systems.

## 1 Introduction

In recent years, advancements in neural networks, particularly the development of large-scale models, have enabled these systems to match or even surpass human performance across various tasks (Devlin et al., 2018; Yang et al., 2019; Liu et al., 2019; Brown et al., 2020; Raffel et al., 2020; He et al., 2022). However, the underlying mechanisms behind their robust generalization abilities, and the exact impact of large-scale data and pretraining on enhancing this capacity, remain open questions. This has led to increased interest in theoretical explanations of neural network generalization, with the concept of low-dimensional representations emerging as a prominent direction of study (Yu et al., 2024; Dai et al., 2023; Chen et al., 2022; Chan et al., 2022; Ansuini et al., 2019).

Recent research suggests that neural networks inherently compress data during processing, with stronger compression often correlating with better generalization performance (Shwartz-Ziv et al., 2018; Ansuini et al., 2019; Recanatesi et al., 2019). This phenomenon has encouraged the design of neural architectures and tasks that enhance a model's ability to learn effective low-dimensional representations (Yu et al., 2024; Chan et al., 2022). While much attention has been given to the benefits of compression, several critical questions remain unanswered: Do neural networks compress different types of data uniformly? If not, how does dynamic compression influence information encoding and generalization across tasks? Addressing these questions is key to understanding the relationship between data complexity, compression and generalization performance in large-scale models.

In this study, we investigate whether neural networks exhibit consistent compression across different data types and tasks. Through an analysis of large models across various datasets

and an examination of neural manifold dimensionality across various tasks, we find that these systems employ a dynamic compression encoding mechanism. Specifically, they apply greater compression to more complex information, ultimately forming uniform low-dimensional representation manifolds.

Furthermore, we establish a theoretical link between low-dimensional representation manifolds and interpolation probability, demonstrating that as the manifold dimension decreases, the probability of interpolation increases. This relationship enhances the system's ability to generalize by representing new data as convex combinations of existing samples. Additionally, we present an upper bound on the generalization error within the interpolation regime, where lower-dimensional representations yield smaller error bounds.

Our contributions can be summarized as follows:

- We analyze the embeddings generated by the pre-trained model CLIP across different datasets, demonstrating that the model compresses complex data to a greater extent than simple data. This supports the hypothesis that large models employ dynamic compression to form uniform low-dimensional representations.
- We investigate the intrinsic dimensions of EEG signals across tasks, revealing that no significant differences are found across tasks, providing evidence of dynamic compression in neural systems.
- We theoretically demonstrate that uniform low-dimensional manifolds enhance interpolation probability, leading to more efficient information encoding and a tighter generalization error bound in low-dimensional spaces.

## 2 Related Works

**Low-dimensional Representation of Neural Network**: Numerous studies have shown that neural networks inherently compress data during processing, and this compression is closely linked to their generalization performance (Yu et al., 2024; Dai et al., 2023; Chen et al., 2022; Chan et al., 2022). The concept of intrinsic dimension has been introduced as a measure of the complexity of the manifold on which data resides. By comparing the intrinsic dimensions of raw data and their representations, the extent of compression achieved by a neural network can be quantified. Research by Ansuini et al. demonstrated that as data progresses through the layers of a neural network, the intrinsic dimensionality of the representation consistently decreases, reflecting the network's ability to compress information (Ansuini et al., 2019). Besides, stronger compression usually correlates with better generalization (Ansuini et al., 2019; Recanatesi et al., 2019). However, these studies primarily focus on the compression of data within individual datasets. A crucial question remains: Do neural networks compress different types of data uniformly, or does the level of compression vary based on the complexity of the input data?

**Low-dimensional Representation of Human Brain**: Similarly, the brain employs compressed representations to efficiently encode information. Studies in the dorsal cortex of awake mice, for example, have shown that a small number of spatiotemporal patterns account for the majority of cortical variability, suggesting that neural representations are inherently low-dimensional (MacDowell & Buschman, 2020). In another study, neurons in the hippocampus were found to use low-dimensional representations to encode spatial and auditory information, further underscoring the functional relevance of low-dimensionality in biological systems (Nieh et al., 2021). Darshan et al. have shown that despite the low-dimensional nature of these neural representations, the nervous system can flexibly adapt to new tasks, adjusting its representations in response to environmental changes (Darshan & Rivkind, 2022). This adaptive learning capability suggests that the brain not only compresses information but also dynamically modifies these compressed representations to support continuous learning. However, it is not yet clear whether the degree of compression in neural systems varies based on the complexity of the tasks being performed. This question is critical to understanding the neural basis of generalization.

**Dimension, Interpolation and Generalization**: Another critical aspect of generalization is the relationship between data dimensionality and interpolation probability. Neural

networks are known to generalize more effectively when performing interpolation, where test samples fall within the convex hull of the training data (Barnard & Wessels, 1992; Haley & Soloway, 1992). However, in high-dimensional spaces, the probability of interpolation decreases dramatically due to data sparsity (Balestriero et al., 2021). Recent studies suggest that neural networks mitigate this issue by compressing data, effectively reducing the dimensionality of their representations and increasing the probability of interpolation (Bárány & Füredi, 1988; Balestriero et al., 2021). Our work extends these findings by demonstrating that dynamic compression in both neural networks and the brain increases interpolation probability, enhancing generalization in both systems.

## 3 Preliminaries and Technical Background

In this section, we provide the theoretical foundation for the analysis presented in the paper. We introduce key concepts such as intrinsic dimension, convex hull, and interpolation probability, which are essential for understanding how low-dimensional representations influence generalization performance in both neural networks and biological systems.

### 3.1 Intrinsic dimension and Ambient Dimension

Let $\mathcal{P} \subset R^N$ represent a set of sample points. We assume that these points lie on a low-dimensional manifold $\mathcal{M} \subset R^N$, where $N$ is the ambient dimension of the space. The ambient dimension $dim(R^N) = N$ refers to the dimension of the surrounding space, while the intrinsic dimension $dim(\mathcal{M}) = d \ll N$ refers to the dimension of the manifold on which the data lies. In essence, the intrinsic dimension quantifies the complexity of the underlying structure of the data.

For example, while neural activity data may be recorded in a high-dimensional space (e.g., from hundreds of electrodes), the underlying complexity of the neural dynamics is often much lower, as reflected by the intrinsic dimension.

### 3.2 Estimation of the intrinsic dimension

To estimate the intrinsic dimension of a manifold, we employ the Maximum Likelihood Estimation (MLE) method proposed by Levina et al. (Levina & Bickel, 2004). This technique relies on the distances between neighboring points in the dataset to compute the manifold's intrinsic dimension.

The intrinsic dimension $\hat{m}_k(x)$ at a point $x$ can be estimated as follows:

$$\hat{m}_k(x) = [\frac{1}{k-1} \sum_{j=1}^{k-1} log \frac{T_k(x)}{T_j(x)}]^{-1}, \tag{1}$$

where $T_j(x)$ denotes the Euclidean distance from point $x$ to its $j^{th}$ nearest neighbor. By averaging these local estimates across all samples, we obtain a global estimate for the intrinsic dimension:

$$\bar{m}_k = \frac{1}{n} \sum_{i=1}^{n} \hat{m}_k(x_i), \tag{2}$$

The parameter $k$ controls the number of neighbors considered when estimating the dimension. A smaller $k$ focuses on a more local perspective, while a larger $k$ captures a more global view of the manifold. By varying $k$ , we can derive a more comprehensive understanding of the manifold's intrinsic dimension.

### 3.3 Convex hull

The convex hull of a set of points is the smallest convex set that contains all the points.

**Definition 1. *Convex Hull*:** *Given a set of points $X = \{x_1, x_2, \ldots, x_n\} \subset \mathbb{R}^d$, the* **convex hull** *of $X$ is defined as:*

$$Conv(X) = \left\{ \sum_{i=1}^{n} \lambda_i x_i \;\middle|\; \lambda_i \geq 0, \sum_{i=1}^{n} \lambda_i = 1 \right\}. \tag{3}$$

### 3.4 Interpolation

Interpolation occurs when a new sample lies within the convex hull of the training data, while extrapolation occurs when the new sample lies outside the convex hull. Formally, we define interpolation probability as follows:

**Definition 2. *Interpolation Probability*:** *Let $X$ be a d-dimensional random vector and $X_1, X_2, \ldots$ be independent copies of $X$. For each $\theta \in \mathcal{R}^d$ and positive integer n, define*

$$p_{n,X}(\theta) := \mathcal{R}(\theta \in conv\{X_1, \ldots, X_n\}), \tag{4}$$

*where $conv\, A := \{\sum_{i=1}^{m} \lambda_i x_i | m \geq 1, x_i \in A, \lambda_i \geq 0, \sum_{i=1}^{m} \lambda_i = 1\}$ denotes the convex hull of a set $A \subset \mathcal{R}^d$.*

## 4 Experiments and Results

To investigate how neural networks and the brain compress different types of information, we conducted two sets of experiments: (1) analyzing the intrinsic dimensions of the pre-trained embedding across various datasets and (2) examining the intrinsic dimensions of EEG signals across different tasks.

### 4.1 Uniform low-dimensional representation in Neural Network

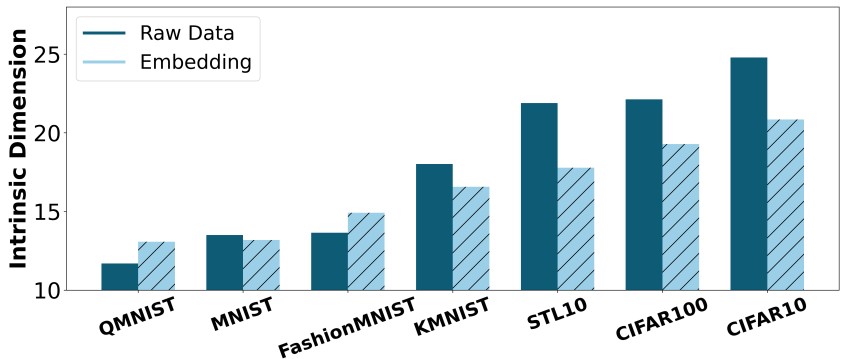

Figure 1: **Intrinsic dimension of raw data and embedding calculated by CLIP.** For complex datasets, the intrinsic dimension of the embeddings significantly decreases compared to the original data. However, for simple datasets, the intrinsic dimension of the embeddings is close to that of the original data.

Previous studies have shown that neural networks compress data into lower-dimensional representations during processing. However, it remains unclear whether this compression is uniform across different types of data or varies dynamically with data complexity. To address this, we used the pre-trained CLIP model (Radford et al., 2021) to analyze seven datasets: QMNIST (Yadav & Bottou), MNIST (LeCun et al., 2010), FashionMNIST (Xiao et al., 2017), KMNIST(Clanuwat et al., 2018), STL10 (Coates et al., 2011), CIFAR10 (Krizhevsky, 2009), and CIFAR100 (Krizhevsky, 2009). The datasets vary in complexity, providing an ideal testbed to examine how neural networks apply dynamic compression.

For each dataset, we computed the intrinsic dimensions of both the original data and the corresponding embeddings generated by the CLIP model. To ensure fair comparisons, all images were resized to 16x32 pixels, matching the embedding space dimensions.

As shown in Figure 1, the intrinsic dimensions of the original data varied considerably across datasets. More complex datasets, such as STL10, CIFAR10, and CIFAR100, exhibited higher intrinsic dimensions, while simpler datasets, like QMNIST, MNIST, and FashionMNIST, showed lower intrinsic dimensions. The CLIP model applied more aggressive compression to the complex datasets, resulting in significantly lower-dimensional embeddings, whereas for the simpler datasets, the compression was less pronounced, with the embeddings' intrinsic dimensions remaining closer to those of the original data.

These findings suggest that neural networks dynamically adjust their compression strategies based on the complexity of the input data, applying stronger compression to more complex datasets. This adaptive compression facilitates the formation of uniform low-dimensional representation manifolds.

## 4.2 Uniform low-dimensional representation in Human Brain

We extended our analysis to biological neural systems to determine whether the brain exhibits similar compression behavior. Specifically, we analyzed the intrinsic dimensions of neural signals (iEEG (Sakakura et al., 2023) and EEG (Wang et al., 2022)) and compared them to environmental sounds (rain, car horns, airplane noises (Piczak, 2015)) and synthetic data (Gaussian noise, uniform noise, sinusoidal waves). Each dataset consisted of 1,000 samples, and all signals were standardized to a consistent ambient dimension (Other technical details are provided in the appendix A).

As summarized in Table 1, the intrinsic dimensions of environmental sounds were significantly higher than those of neural signals. This indicates that the brain compresses external information much more efficiently than other types of signals, reflecting the efficient low-dimensional encoding inherent to neural systems.

Table 1: Intrinsic dimension of different data modalities

| Type | Data modality | k=10 | k=20 | k=30 | k=40 | k=50 |
|---|---|---|---|---|---|---|
| **Neural Signal** | iEEG signals | 8.973 | 7.828 | 7.283 | 6.904 | 6.655 |
| | EEG signals | 11.157 | 9.646 | 8.931 | 8.465 | 8.166 |
| **Ambient Sounds** | Rain | 63.712 | 56.880 | 53.863 | 51.770 | 50.230 |
| | Car horn | 25.985 | 23.735 | 23.224 | 22.783 | 22.387 |
| | Airplane | 49.041 | 45.547 | 44.712 | 44.121 | 43.504 |
| | Church bells | 44.832 | 43.617 | 43.490 | 43.351 | 43.131 |
| **Synthetic Data** | Gaussian noise | 75.635 | 67.507 | 63.963 | 61.497 | 59.492 |
| | Uniform noise | 77.388 | 67.649 | 63.649 | 61.317 | 59.518 |
| | Sinusoidal waves | 4.041 | 6.485 | 8.723 | 10.887 | 12.991 |

Next, we explored the variability of EEG intrinsic dimensions across different tasks, including both resting-state and task-specific conditions, with a particular focus on eyes-open (EO) and eyes-closed (EC) states. The results are illustrated in Figure 2.

In the EO state, the intrinsic dimension of EEG was significantly higher compared to the EC state, indicating that the brain's encoding complexity increases when processing visual information. However, within the same state (either EO or EC), there were no significant differences in intrinsic dimension between resting and task-specific conditions (Results of statistical analysis and intrinsic dimension analysis with other algorithms are provided in the appendix B). This suggests that while the brain adjusts its compression based on sensory input, the overall complexity of neural representations remains stable across tasks.

These findings demonstrate that the brain, like neural networks, employs dynamic compression to create uniform low-dimensional manifolds for efficient encoding of information.

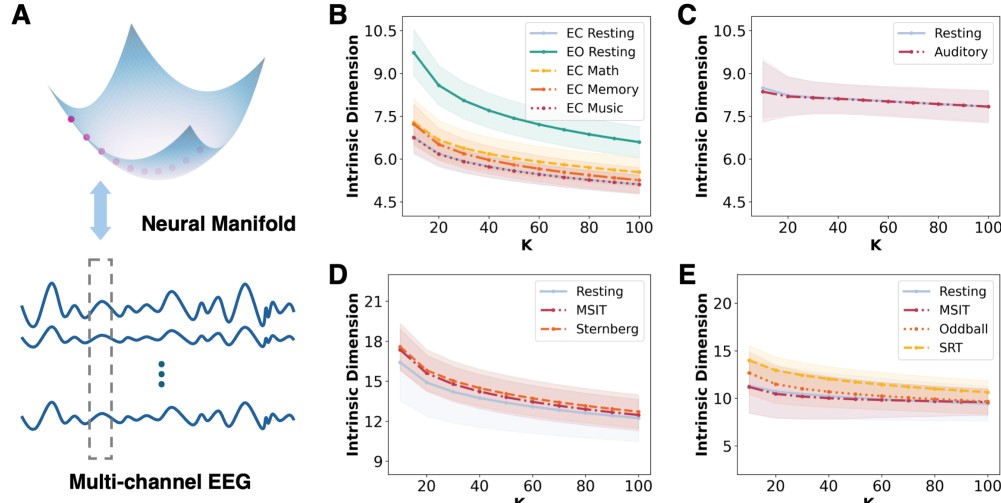

Figure 2: **Intrinsic dimension analysis of EEG under different tasks**. (A) Schematic illustration of intrinsic dimension computation for multi-channel EEG. Each time point's multi-channel EEG data is treated as a high-dimensional vector, and different time points form discrete samples in this high-dimensional space. These samples are used to compute the intrinsic dimension of the neural manifold. (B) Intrinsic dimension comparison between eyes-closed (EC) and eyes-open (EO) resting states and task states. The intrinsic dimension of EO resting state is significantly higher than that of EC resting state and EC task states. However, there is no significant difference between EC resting state and EC task states. (C-E) Intrinsic dimension analysis of EO resting state and various task states. No significant differences were observed between the EO resting state and task states in any of the comparisons.

## 5    Impact of Manifold Dimension on Interpolation Probability

The above analysis emphasizes that both neural networks and the brain learn uniform low-dimensional representation manifolds through dynamic compression encoding. Next, we explore theoretically the role of low-dimensional manifolds in information representation. Here, we analyze from the perspective of interpolation, noting that as data dimensionality decreases, interpolation probability increases, which means new samples are more likely to be represented as convex combinations of existing samples.

**Theorem 5.1** ((Bárány & Füredi, 1988)). *Given a d-dimensional dataset $X \triangleq x_1, ..., x_N$ with i.i.d. samples uniformly drawn from a hyperball, the probability that a new sample x is in the interpolation regime exhibits the following asymptotic behavior:*

$$lim_{d\to\infty} p(x \in Conv(X)) = \begin{cases} 1 \Leftrightarrow N > d^{-1}2^{d/2} \\ 0 \Leftrightarrow N < d^{-1}2^{d/2} \end{cases} \quad (5)$$

**Theorem 5.2** ((Kabluchko & Zaporozhets, 2020)). *Let X consist of N i.i.d. d-dimensional samples from $\mathbb{N}(0, I_d)$ with $N \geq d+1$, then for every $\sigma \geq 0$ the probability that a new sample $x \sim \mathbb{N}(0, \sigma^2 Id)$ is in extrapolation regime is given by*

$$p(x \notin Conv(X)) = 2(b_{N,d-1}(\sigma^2) + b_{N,d-3}(\sigma^2) + ...) \quad (6)$$

*with*

$$b_{n,k}(\sigma^2) = \binom{n}{k} g_k(-\frac{\sigma^2}{1+k\sigma^2}) g_{n-k}(\frac{\sigma^2}{1+k\sigma^2}), \ g_n(r) = \frac{1}{\sqrt{2\pi}} \int_{-\infty}^{\infty} \Phi^n(\sqrt{r}x) e^{-x^2/2} dx$$

*where $\sqrt{r} = i\sqrt{-r}$ if $r < 0$ and $b_{N,k} = 0$ for $k \notin \{0, 1, ..., N\}$.*

Theorem 5.1 indicates that as dimensionality increases, the convex hull struggles to cover the entire data space, causing a significant drop in interpolation probability. In high-dimensional

spaces, maintaining a high interpolation probability requires an exponential increase in data size. In contrast, in low-dimensional spaces, data points are denser, making it easier for the convex hull to cover new samples, resulting in a higher interpolation probability.

Theorem 5.2 quantitatively describes the probability of extrapolation in high-dimensional spaces. As dimensionality increases, the likelihood of extrapolation rises, and interpolation probability decreases.

Low-dimensional spaces offer higher interpolation probabilities, enabling effective generalization with fewer data points. In contrast, high-dimensional spaces require significantly more data to achieve similar results, highlighting the value of low-dimensional representations in neural networks. This principle also applies to biological neural systems, where increased interpolation probabilities improve information encoding efficiency. Our EEG analysis indicates that neural representations adapt to task difficulty, allowing the brain to generalize quickly from past experiences. This dynamic coding strategy supports cognitive flexibility and decision-making while minimizing computational demands.

However, if the data distribution is non-uniform, the advantages of low-dimensional representations may be diminished. Sparse regions in the data space can reduce interpolation effectiveness, leading to increased extrapolation errors. Therefore, both low dimensionality and uniformity of the representation manifold are essential. Uniform distribution enhances interpolation probability, enabling better generalization and improving encoding efficiency. For optimal performance, it is crucial to ensure that both neural networks and biological systems maintain low-dimensional, uniformly distributed representations.

## 6 Existence of Generalization Error Bound in the Interpolation Regime and the Impact of Dimension

Low-dimensional representations can increase interpolation probability, thereby enhancing the efficiency of information encoding in systems. In this section, we further theoretically demonstrate that, within the interpolation regime, neural networks have a generalization error upper bound, which decreases as the dimensionality becomes smaller.

**Theorem 6.1.** *Let $\ell(y, x, \theta)$ be a loss function that is Lipschitz continuous with respect to both $x \in \mathbb{R}^d$ and $y \in \mathbb{R}^k$, with Lipschitz constant $L$. Assume that the input data $x$ and output data $y$ are bounded such that $\|x - x'\| \leq D_x$ and $\|y - y'\| \leq D_y$ for all $x, x'$ and $y, y'$. Let $\hat{L}(\theta, D)$ be the empirical loss over a dataset $D = \{(x_i, y_i)\}_{i=1}^n$, and let $L(\theta)$ be the expected loss over the data distribution $v$. Then, for any $\epsilon > 0$, the following bound holds:*

$$P\left(\left|\hat{L}(\theta, D) - L(\theta)\right| \geq \epsilon\right) \leq 2\exp\left(-\frac{2n\epsilon^2}{L^2(D_x + D_y)^2}\right). \tag{7}$$

*Furthermore, if the Lipschitz constant $L$ and the data diameters $D_x$ and $D_y$ scale with the dimension $d$ as $L = C_L\sqrt{d}$ and $D_x = C_x\sqrt{d}$, while $D_y$ is constant, then the bound becomes:*

$$P\left(\left|\hat{L}(\theta, D) - L(\theta)\right| \geq \epsilon\right) \leq 2\exp\left(-\frac{2n\epsilon^2}{C^2 d^2}\right), \tag{8}$$

*where $C = C_L(C_x + C_y/\sqrt{d})$ and for large $d$, $C \approx C_L C_x$. This shows that the generalization error bound becomes tighter as the dimension $d$ decrease.*

This theorem highlights the critical role of dimension in generalization performance. While the dimension of raw data remains fixed, we can shift the focus from the dimension of raw data to the dimension of the learned representations. In this context, lower representation dimension leads to better generalization performance.

# 7 Discussion and Conclusion

In this study, we demonstrated that dynamic compression mechanisms in both neural networks and the brain lead to the formation of uniform low-dimensional representation manifold. This manifolds plays a pivotal role in enhancing interpolation probability, which, in turn, contributes to improved generalization capabilities. This resemblance in information processing strategies between artificial neural networks and biological neural systems underscores the universality of efficient low-dimensional encoding across intelligent systems.

While our research provides valuable insights into the interplay between data complexity, compression and generalization, it also has several limitations that warrant further exploration:

- **Expansion to Other Neural Signal**: Although this work focuses on EEG and iEEG data, future studies should investigate whether similar compression patterns are observed in other neural modalities. We focused on EEG and iEEG due to their sufficient temporal resolution, which allows each moment's multi-channel information to be treated as a vector, with different moments serving as different samples. Current dimensionality estimation algorithms produce reliable results only when the number of samples exceeds the dimensionality (Levina & Bickel, 2004). For signals like fMRI, which have high spatial but low temporal resolution (Goense et al., 2016), the algorithm either becomes inaccurate or can only be applied to local brain regions. Therefore, to extend this analysis to other neural modalities like fMRI or MEG, improvements in dimensionality estimation algorithms are necessary.

- **Task Complexity and Dimensionality**: Our results indicate that the intrinsic dimension of neural manifold remains stable across tasks of varying complexity, such as resting states and task-specific conditions. However, further research is needed to assess whether more cognitively demanding tasks, which involve higher-order reasoning or abstract thought, could alter the brain's compression dynamics (Kool et al., 2010; Botvinick & Rosen, 2009; Kraus et al., 2023). Investigating how the brain adapts its encoding strategies based on task complexity would deepen our understanding of cognitive flexibility.

- **Optimization of Compression Mechanisms in Neural Networks**: While we have shown that neural networks employ dynamic compression to adapt to varying data complexity, more work is required to optimize these mechanisms. Specifically, future research could explore how incorporating architectural modifications such as attention mechanisms or sparsity constraints could further enhance a model' s ability to generalize across different domains. This could lead to more robust AI systems that better mimic the flexibility of biological systems.

- **Impact of Non-Uniform Manifolds**: Our analysis focused on uniform low-dimensional manifolds, yet real-world data often exhibit non-uniform distributions (Crovella et al., 1998). Exploring the impact of non-uniform manifolds on interpolation probability and generalization would provide a more realistic understanding of how both biological and artificial systems handle complex, unevenly distributed data.

- **Handling of Out-of-Distribution Data**: Our analysis focused on interpolation within the convex hull of training data. However, a key limitation is the treatment of out-of-distribution (OOD) data, which may lies outside the convex hull (Liu et al., 2021). Neural networks may struggle to generalize when confronted with OOD data, leading to higher error rates and reduced performance. Future studies should investigate how neural networks can be enhanced to handle such data more effectively, either through architectural innovations or training strategies that improve extrapolation capabilities. Understanding how biological systems manage OOD information could also provide valuable insights.

In conclusion, dynamic compression strategies contribute significantly to the formation of uniform low-dimensional representation manifolds, a key factor in both neural and artificial systems' ability to generalize effectively. This work highlights the parallels between biological

information processing and AI, offering new avenues for the development of more efficient models. Our findings lay a foundation for future research aimed at optimizing compression mechanisms and exploring the broader implications of low-dimensional representations across various domains.

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

## A   Technical Details Summary

To ensure robust intrinsic dimension analysis, we primarily utilized the skdim toolkit from scikit-learn (Bac et al., 2021). For our calculations, we applied the Maximum Likelihood Estimation (skdim.id.MLE()), Method Of Moments (skdim.id.MOM()), and Tight Local intrinsic dimensionality Estimator (skdim.id.TLE()) algorithms. The key hyperparameter for these functions is $k$, representing the number of nearest neighbors. We experimented with 10 different hyperparameter settings, ranging from $k = 10$ to $k = 100$, to estimate the intrinsic dimension across datasets. EEG data preprocessing was conducted using the MNE toolkit, where all EEG signals were resampled to 250 Hz, band-pass filtered between 1-80 Hz, and normalized using z-score scaling.

## B   Statistical Analysis and Algorithm Validation of Section 4.2

To evaluate the statistical significance of intrinsic dimension differences across various sensory and task conditions, we employed the Wilcoxon signed-rank test as in Figure 3. Our analysis revealed significant differences between the eyes-open (EO) and eyes-closed (EC) states, suggesting that sensory conditions notably impact the complexity of neural representations, as measured by intrinsic dimension. However, within each sensory condition—whether in the EO or EC state—no significant differences were found between resting-state and task-specific conditions (e.g., resting vs. memory tasks, or resting vs. auditory tasks).

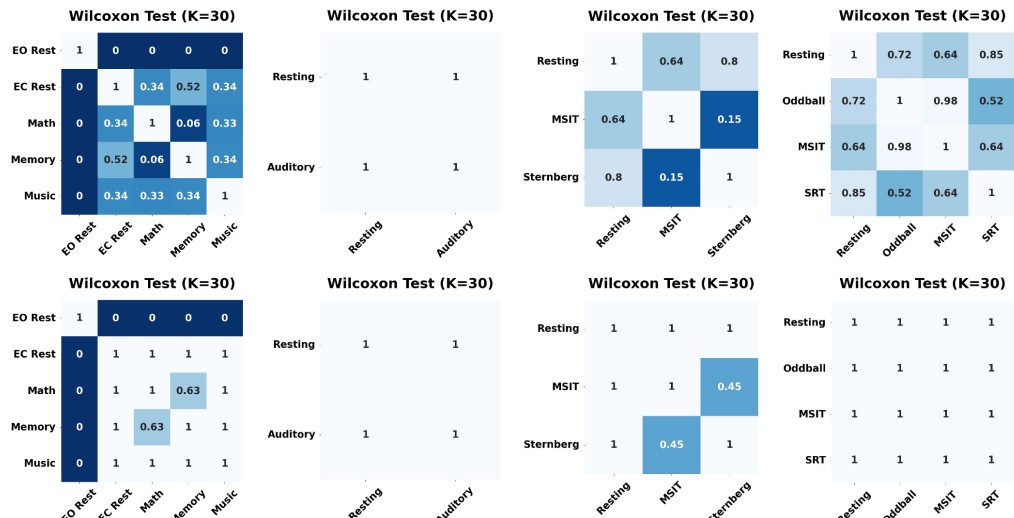

Figure 3: **Results of Wilcoxon signed-rank test before and after Bonferroni correction.** The first row presents the original results of the Wilcoxon signed-rank test, while the second row shows the results after applying the Bonferroni correction.

This lack of differentiation indicates that, for both EO and EC states, the brain maintains a consistent level of intrinsic dimension across a range of cognitive tasks. Whether at rest or engaged in different tasks, the neural manifold exhibits stability in its dimensional complexity, suggesting that task-related processing does not induce substantial changes in the brain's overall representational structure, at least at the level of intrinsic dimensionality.

To ensure the robustness of these findings, we applied Bonferroni correction for multiple comparisons. The results remained consistent after correction, further reinforcing the conclusion that intrinsic dimensionality is preserved across tasks, regardless of whether the brain is in an active cognitive state or a resting state.

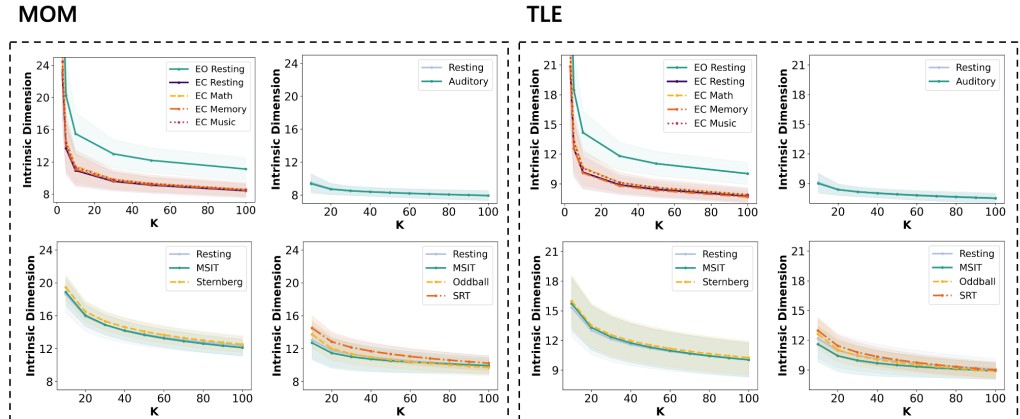

Figure 4: **Intrinsic dimensionality across various states estimated with the MOM and TLE algorithm.** The intrinsic dimension estimates obtained using the MOM and TLE algorithms were consistent with those from the MLE algorithm. The only significant difference observed was between the EO and EC conditions. Regardless of whether the brain was engaged in a resting state or a task, the intrinsic dimension of neural activity remained stable as long as the sensory condition (EO or EC) was maintained.

To further validate the accuracy and stability of our dimensionality analysis, we recalculated the intrinsic dimensions using two additional algorithms: MOM and TLE algorithms. The results, shown in Figure 4, were consistent with those obtained from the MLE algorithm, reinforcing the reliability and robustness of our methods across various computational approaches.

## C  PROOF OF THEOREM 6.1

**Definitions   Empirical Loss**:

$$\hat{L}(\theta, D) = \frac{1}{n}\sum_{i=1}^{n}\ell(y_i, x_i, \theta),$$

where $D = \{(x_i, y_i)\}_{i=1}^{n}$ is the dataset.

**Expected Loss**:

$$L(\theta) = \mathbb{E}_{(x,y)\sim v}[\ell(y, x, \theta)],$$

where $v$ is the data distribution.

**Objective**   Our goal is to bound the probability:

$$P\left(\left|\hat{L}(\theta, D) - L(\theta)\right| \geq \epsilon\right).$$

**Step 1: McDiarmid's Inequality**   McDiarmid's inequality states that if $X_1, X_2, \ldots, X_n$ are independent random variables taking values in a set $A$, and the function $f : A^n \rightarrow \mathbb{R}$ satisfies the bounded differences condition:

$$\sup_{x_1,\ldots,x_n,x_i'} |f(x_1, \ldots, x_i, \ldots, x_n) - f(x_1, \ldots, x_i', \ldots, x_n)| \leq c_i,$$

then for all $\epsilon > 0$:

$$P\left(f(X) - \mathbb{E}[f(X)] \geq \epsilon\right) \leq \exp\left(-\frac{2\epsilon^2}{\sum_{i=1}^{n} c_i^2}\right).$$

**Step 2: Bounded Differences Condition**  We need to verify the bounded differences condition for the empirical loss function $\hat{L}(\theta, D)$ when one sample $(x_i, y_i)$ is replaced by another $(x_i', y_i')$.

Define:

$$\Delta_i = \left| \hat{L}(\theta, D) - \hat{L}(\theta, D_i') \right|,$$

where $D_i'$ is the dataset $D$ with the $i$-th sample replaced by $(x_i', y_i')$.

Compute $\Delta_i$:

$$\Delta_i = \left| \frac{1}{n} \left( \ell(y_i, x_i, \theta) - \ell(y_i', x_i', \theta) \right) \right|.$$

**Step 3: Applying Lipschitz Continuity**  By the Lipschitz continuity of $\ell$, we have:

$$|\ell(y_i, x_i, \theta) - \ell(y_i', x_i', \theta)| \leq L \left( \|x_i - x_i'\| + \|y_i - y_i'\| \right).$$

Therefore,

$$\Delta_i \leq \frac{L}{n} \left( \|x_i - x_i'\| + \|y_i - y_i'\| \right).$$

Using the boundedness of the data:

$$\|x_i - x_i'\| \leq D_x, \quad \|y_i - y_i'\| \leq D_y,$$

so we have:

$$\Delta_i \leq \frac{L}{n}(D_x + D_y) = c_i.$$

**Step 4: Calculating the Sum of $c_i^2$**  Since $c_i = \frac{L}{n}(D_x + D_y)$ for all $i$, we have:

$$\sum_{i=1}^{n} c_i^2 = nc_i^2 = n\left( \frac{L}{n}(D_x + D_y) \right)^2 = \frac{L^2(D_x + D_y)^2}{n}.$$

**Step 5: Applying McDiarmid's Inequality**  Applying McDiarmid's inequality:

$$P\left( \hat{L}(\theta, D) - \mathbb{E}[\hat{L}(\theta, D)] \geq \epsilon \right) \leq \exp\left( -\frac{2\epsilon^2}{\sum_{i=1}^{n} c_i^2} \right) = \exp\left( -\frac{2n\epsilon^2}{L^2(D_x + D_y)^2} \right).$$

Similarly, for the lower tail:

$$P\left( \hat{L}(\theta, D) - \mathbb{E}[\hat{L}(\theta, D)] \leq -\epsilon \right) \leq \exp\left( -\frac{2n\epsilon^2}{L^2(D_x + D_y)^2} \right).$$

Combining both tails:

$$P\left( \left| \hat{L}(\theta, D) - \mathbb{E}[\hat{L}(\theta, D)] \right| \geq \epsilon \right) \leq 2\exp\left( -\frac{2n\epsilon^2}{L^2(D_x + D_y)^2} \right).$$

**Step 6: Connecting to Expected Loss**  Since samples are independent and identically distributed (i.i.d.) from distribution $v$, we have:

$$\mathbb{E}[\hat{L}(\theta, D)] = L(\theta).$$

Therefore:

$$P\left( \left| \hat{L}(\theta, D) - L(\theta) \right| \geq \epsilon \right) \leq 2\exp\left( -\frac{2n\epsilon^2}{L^2(D_x + D_y)^2} \right).$$

This proves the first part of the theorem.

**Step 7: Dependence on Dimension** $d$  Assume the following scaling with dimension $d$:

1. **Lipschitz Constant** $L$:
$$L = C_L \sqrt{d},$$
where $C_L$ is a constant independent of $d$.

2. **Data Diameter** $D_x$:
$$D_x = C_x \sqrt{d},$$
where $C_x$ is a constant.

3. **Data Diameter** $D_y$: For simplicity, assume $D_y$ is constant (i.e., the dimension of $y$ does not grow with $d$).

**Step 8: Substituting into the Bound**  Compute the denominator in the exponent:
$$L^2 (D_x + D_y)^2 = (C_L \sqrt{d})^2 (C_x \sqrt{d} + D_y)^2 = C_L^2 d (C_x \sqrt{d} + D_y)^2.$$

For large $d$, $C_x \sqrt{d}$ dominates $D_y$, so:
$$C_x \sqrt{d} + D_y \approx C_x \sqrt{d}.$$

Thus,
$$L^2 (D_x + D_y)^2 \approx C_L^2 d (C_x \sqrt{d})^2 = C_L^2 d (C_x^2 d) = C_L^2 C_x^2 d^2.$$

Therefore, the bound becomes:
$$P\left( \left| \hat{L}(\theta, D) - L(\theta) \right| \geq \epsilon \right) \leq 2 \exp\left( -\frac{2n\epsilon^2}{C_L^2 C_x^2 d^2} \right).$$

Let $C = C_L C_x$, so:
$$P\left( \left| \hat{L}(\theta, D) - L(\theta) \right| \geq \epsilon \right) \leq 2 \exp\left( -\frac{2n\epsilon^2}{C^2 d^2} \right).$$

This proves the second part of the theorem.

**Conclusion**  The bound on the generalization error becomes tighter as the dimension $d$ decreases, specifically due to the $d^2$ term in the denominator of the exponent. This indicates that in lower-dimensional spaces, fewer samples $n$ are required to ensure that the empirical loss $\hat{L}(\theta, D)$ closely approximates the expected loss $L(\theta)$. Therefore, reducing the dimensionality of the input data can significantly improve generalization performance and reduce the risk of overfitting, highlighting the importance of low-dimensional representation for generalization.