# OpenReview forum: "Dynamic Compression Strategies for Uniform Low-Dimensional Representations in Human Brain and Neural Network"
_ICLR.cc/2025/Conference — ICLR 2025 Conference Withdrawn Submission_

### Official Review · Reviewer_RZXt · 2024-10-28

**Soundness:** 2
**Presentation:** 2
**Contribution:** 2
**Rating:** 3
**Confidence:** 5

**Summary:**

**Summary of Contribution**:
The paper proposes that both artificial neural networks and the human brain employ dynamic compression mechanisms to form low-dimensional data representations, enhancing generalization and interpolation. By analyzing intrinsic dimensionality using the MLE method by Levina & Bickel, 2004 in CLIP embeddings and EEG signals, the study claims that both artificial and biological neural networks dynamics adjust the intrinsic dimension of internal representation based on the intrinsic dimension of input data, using higher internal dimension for input data with high intrinsic dimension. The paper show some theoretical results on interpolation probability from Barany & Furedi, 1988 and Kabluchko & Zaporozhets, 2020. Finally the authors show that generalization error bounds tightens as embedding dimension decreases.

**Strengths:**

#### **Strong Points**
1. The idea to look at compression of both biological and artificial neural network is novel.

**Weaknesses:**

#### **Weak Points**
1. The authors should justify this juxtaposition of results on EEG signals and CLIP embeddings. It makes more sense to use neural pixel recording data, for example, from the perspective of system identification.
2. The theory contribution of this paper is weak for the following reasons: 1) the result on interpolation probability is from past work and does not explain why “dynamic” compression is needed. 2) For theorem 6.1, what is actually proved is that if the data is concentrated enough, i.e. (Dx + Dy)^2 is small enough, then there is high probability that the generalization gap is small, which is obvious given the Lipschitz property of the loss function. The handwavy argument that Dx and Dy scale with the dimension should be justified.
3. While the authors claim that dynamic compression is beneficial for generalization, no empirical result on either artificial or biological networks is shown regarding the generalization performance. The authors should think of a unified framework to evaluate generalization performance on difference datasets.
4. The concept of dynamic compression is rather qualitative in the way that it is introduced in the paper. The result in the figure should be more thoroughly explored. For example, while MNIST and FashionMNIST has the same raw data distribution, but FashionMNIST’s embedding intrinsic dimension is higher (than the raw data even).

**Questions:**

See weaknesses. My suggestion is to 1) More explicitly build the connection between the result of artificial and biological networks, e.g. is this consistency between artificial and biological networks surprising? *why*?  2) establish mathematical theory on *dynamic* compression rather than compression.

---

### Official Review · Reviewer_HJ4y · 2024-10-29

**Soundness:** 2
**Presentation:** 3
**Contribution:** 2
**Rating:** 3
**Confidence:** 4

**Summary:**

In this paper, the authors use an intrinsic dimensionality estimator based on maximum likelihood estimate to estimate the dimensionality of neural network represenations of different datasets, along with that of brain activity measured by EEG under various conditions. From these measurements, the authors argue that both artificial neural networks and biological ones emply a dynamic compression scheme, whereby complex data is compressed more than simple one. They then argue that is is better for generalization and present a theoretical result based oninterpolation bounds.

**Strengths:**

The paper investigates an interesting question: the link between intrinsic dimensionality of representations and the generalization properties of neural networks. To support this investigation, the approximate intrinsic dimension of representations in both artificial neural networks and in the brain. They also offer a theoretical result outliuning generalization property and it's link to dimension.

**Weaknesses:**

In its current form, the paper lacks in a few ways. The biggest problem, in my view, is the disconnect between the theoretical result and the empirical results. Indeed, the authors first measure the intrinsic dimension of representations in pre-trained networks and in the brain (based on EEG recordings). Using these measurements, they argue that neural networks employ a dynamic compression scheme which adapts the level of dimensionality reduction in latent representations based on the complexity of the data itself. They then present a theorem that prdicts generalization error bounds based on dimensionality of represenations. There are no empirical validation of this bound, and no clear link with the mesurements made, other than measuring dimensionality itself. It is then puzzling to see the link between the two contributions in this paper. The relation "neural networks compress data" + "compressed data helps generalization when doing interpolation" is circumstantial in this case. The authors do not directly test generalization and actually only measure dimensionality in one, pre-trained model. Furthermore, there are some questions about the dimensionality measurements and the comparisons presented. See questions below.

**Questions:**

1. Intrinsic dimensionality of data points can be estimated in a nbumber of ways. For example, using the participation ratio (based on the grammian matrix singular values), correllation dimensions, box counting, etc. In this paper, the authors only use one measurement based on maximum likelihood estimate. This method is known to have some caveats, including sensitivity to outliers with small k, and loss of manifold resolution in high k. Have the authors validated their findings with other easily computable dimensionality estimators? Crucially, this would help better interpret the distinct comparions between dimensionality of data, and that of representations. This brings me to a second point.

2. When comparing dimensionality of data with that of neural network representations (e.g. Fig.1) it is not clear how one should interpret the two quantities as they measure dimensions in distinct spaces. What was the metric used in computing the distance between two points in image space? Different metric can be used there, is the measure of dimensionality used invariant to this change of metric? Different controls would help shed light on this. For example, computing the dimensionality of the dataset with shuffled pixels would indicate the level of structure in the image. It would also be informative to mesasure the dimensionality of images filled with uncorrelated white noise, and measure the representation dimensionality of the CLIP model for these inptus. Would this be different from the other datasets?

3. The dim measurement of EEG data in comparison to different stimuli in the world, when the EEG data is in no way linked to the perception of such stimuli, is puzzling. What should we conclude in from Table 1? The neural signal at rest has lower dim than the sound of rain on average? Without input-output comparison, there is appers to be little value in comparing intrinsic dimensions, unless I am missing something.

4. In Fig.2, the only significant difference between intrinsic dim estimate from EEG is between eyes closed or eyes opened states. It is well known that this is the case, given that sensory stimuli engage sensory cortices that are clse to the periphery and therfore drive signals that can be recorded by EEG. However, there is no significant difference between the EGG activity during different tasks. Despite this, the authors write "Our EEG analysis indicates that neural representations adapt to task diﬃculty, allowing the brain to generalize
quickly from past experiences" (line 335). This statement appears to be in direct contradiction with the results shown in Fig.2. Maybe I am misunderstanding, I would apprecite some clarification by the authors.

5. The theoretical result presented is interesting and appears sound and informative. However, I see no link between this result, which links generalization under some loss function and intrinsic dimension of a representation, and the data presented eralier in the paper. It would be interesting to see some experimental validation of this generalization bound, perhaps with some tractable regression task where dimensionality is controlled?

---

### Official Review · Reviewer_dL5t · 2024-11-01

**Soundness:** 3
**Presentation:** 3
**Contribution:** 1
**Rating:** 3
**Confidence:** 3

**Summary:**

The paper conducts an experimental study that  measures the intrinsic dimensionality (ID) of
 1. various natural and synthetic signals,
 2. CLIP embedding of several image datasets and
 3. EEG measurements of neural activity.

Furthermore the author's provide a new (to the best of my knowledge) upper bound on the generalization error for certain types of loss functions given that the new sample produces an embedding that lies within the convex hull of the training dataset embeddings. The author's use this result (along with existing work on the curse of dimensionality that the probability of interpolation is vanishingly small for finite samples in high dimensional spaces) to suggest a reason that network embeddings generally have lower ID than their corresponding input datasets (to increase the probability of generalizing via interpolation).

**Strengths:**

- The subject of the study (how input dimensionality is related to the dimensionality of representations in neural networks) is interesting.
- The exposition (description of intrinsic dimensionality, interpolation probability, etc.) is well written and informative.
- The theoretical contribution (bounding generalization error for a class of loss functions) given uniform representations as a function of dimensionality, seems both interesting and sound.

**Weaknesses:**

- To my eye, the empirical results of Figure 1 don't strongly (or at least, clearly)  support the "uniform low dimensional representation" hypothesis. Firstly, for particularly simple datasets the ID of embeddings can actually be larger than that of the input data. Secondly, the ID does not seem to be uniform across different input datasets and indeed seems to grow at a similar rate to input ID for many of the considered datasets (i.e. in the rightmost four columns). Is the argument in fact that ID is reduced at a uniform rate across different input datasets?

- Only one network (CLIP) is considered, and it is not necessarily obvious that similar observations would be seen in i.e. different architectures or for networks trained using different objectives.

- Section 4.2 is lacking sufficient detail to evaluate the claims:
  - Are the neural signal rows in Table 1 related to the measurements analyzed in Figure 2? Do these represent dimensionalities after aggregating across conditions?
  - Are the natural and synthetic audio signals related to the task conditions referenced in Figure 2? Furthermore, what task are participants performing?
 - Is the takeaway from Table 1 supposed to be that EEG measurements are lower dimensional than natural or synthetic signals? Even if this is the case, this does not constitute sufficient evidence of the claim that the brain prefers "uniform low dimensional representations." For one the EEG signal is at best an incomplete measurement of the brain's representation of these signals. For example, how would the dimensionality change if the number of EEG channels was increased, what if we could perfectly record all of the neural activity of each unit in auditory cortex?

**Questions:**

- In lines 215-216 the author's mention that all datasets are rescaled to 16x32. While I can appreciate that this is in some sense makes the ID comparisons fair, I am not sure it is the right choice. Higher resolution images can contain more complicated signals than their downscaled counterparts (simply because they can contain higher spatial frequencies), and thus the dimensions eliminated by resizing may in fact be meaningful. Furthermore it is unclear whether the inputs were rescaled before they were fed into the CLIP network? If not, this would seem to actually "unfair" in some sense.

- It would be very interesting if the author's could shed some light on how "adaptive compression" is achieved by the CLIP network. For one, are any dimensions of the representation strongly preserved across datasets? The "holy grail" would be to shed light on the mechanisms facilitating such a strategy, though this is a tall task and I admit I do not have any immediate suggestion on how to do so.

- In the paragraph beginning on line 339 the author's discuss the importance of uniformity in confluence with low-dimensionality. Do any of the empirical experiments shed light on the relative uniformity of either artificial or biological representations? If not, this could perhaps be an interesting new direction for observational work.

---

### Official Review · Reviewer_QyyY · 2024-11-04

**Soundness:** 1
**Presentation:** 2
**Contribution:** 1
**Rating:** 3
**Confidence:** 4

**Summary:**

In the work, the relation between intrinsic dimensionality and the origin of data is explored.
It is argued that the gap between the intrinsic dimensionality of the original (uncompressed) data and the corresponding CLIP embeddings
increases with the complexity of the dataset.
A somewhat similar pattern is observed when analysing signals from a human brain:
the results reveal that the intrinsic dimensionality of EEG recordings is higher
when the subject's eyes are open and lower when they are closed.
Some possible effects of lower intrinsic dimensionality on the generalization performance of neural networks are also discussed.

**Strengths:**

1. The article touches on an interesting topic
   and provides some ground for a discussion on similarities between biological and artificial neural networks.
1. The statistical analysis of the results obtained from brain EEG seems rigorous.

**Weaknesses:**

1. The experimental setup and corresponding conclusions regarding artificial NNs are questionable.

   Firstly, there are other equally feasible explanations of the phenomenon observed in Figure 1,
   which should be explored (and negated):
   - As the CLIP model is pretrained on a separate dataset,
     there may be a fixed set of features which this NN is trained to extract.
     Thus, when the NN is fed with simple datasets, the latent dimensions corresponding to the missing features become degenerate
     or even hallucinatory (which might explain the unexpected increase in the dimensionality when embedding "simple" datasets).

     One can even imagine a situation, in which the embedder network is pretrained on a "simple" dataset,
     thus yielding degenerate embeddings for complex datasets and reversing the trend from Figure 1
     (i.e., the embeddings corresponding to complex datasets would be of lower intrinsic dimensionality
     compared to the "simple" counterparts).

     To negate this hypothesis, one should consider using different models pretrained on different datasets with varying complexity
     (or even on the combined dataset).
   - Networks pretrained on one dataset might compress other datasets suboptimally.

     To negate this hypothesis, one should consider training CLIP (or other self-supervised, unsupervised or even supervised models)
     on each dataset separately.
   - The intrinsic dimensionality estimator might become unreliable when the input data reaches certain level of complexity,
     yielding higher estimates for unstructured data given the same ground truth intrinsic dimensionality.

     This can be negated through providing elaborate statistical analysis and confidence intervals.

   Secondly, as the reliability of the intrinsic dimensionality estimator can be questioned,
   the work would benefit greatly from synthetic tests with tractable intrinsic dimensionality
   (can be achieved via using synthetic manifolds).

   Finally, additional details regarding the CLIP model (e.g., the dataset it was pretrained on) should be provided.
1. Using single slices from EEG may lead to loosing temporal structure of the underlying data
   with the corresponding additional level of complexity.
   Other methods of sampling (using several slices, embedding the whole sequence via a NN, etc.) should be considered.
1. Downscaling the images in Section 4.1 can also severely affect the intrinsic dimensionality.
   Additional results for uncompressed images would certainly improve this section.
1. The work would benefit greatly from improving the structure.
   For example, the current division into subsections is too verbose, with Section 3.3 being dedicated to a single definition.
   Please, consider merging sections and/or using the `\paragraph{Paragraph name}` command.
   The definition of a convex hull is provided twice: Definition 1 and line 177.
   The connection between Theorems 5.1-5.2 and 6.1 remains somewhat vague and requires an additional discussion.
   For example, a lemma can be provided that formally connects the results in question.
1. The choice of $C = C_L \sqrt{d}$ and $D_x = C_x \sqrt{d}$ in Theorem 6 seems arbitrary.

**Questions:**

1. As the analysis of empirical risks in various Lipschitz settings is a well-explored topic,
   I kindly ask the authors to provide a corresponding overview and pose their Theorem 6.1 against existing results.
1. What data was used to pretrain the CLIP model?
   What transformations are applied to images of different sizes and numbers of channels before feeding them into CLIP?

**Minor issues and typos:**

1. The "hat" is misplaced on lines 142--143 and in Equation (1): `\hat{m}_k` might be preferable to `\hat{m_k}`.
1. Missing `\left`/`\right` commands in Equation (1).
   Similar issues in Theorem 5.2.
1. Using `\log` instead of `log` might be preferable for readability in Equation (1).
   Similarly, `\lim` instead of `lim` in Equation (5).
1. In Equation (4), should it be $P$ instead of $\mathcal{R}$?
1. One may consider using `\DeclareMathOperator{\conv}{\mathrm{Conv}}` to improve the readability in theorems,
   where the font is italic.
1. Double parentheses in Theorems 5.1 and 5.2 (lines 307 and 313).
   One may consider using `\citealt{reference}`.

---

### Note · Authors · 2024-11-23

I have read and agree with the venue's withdrawal policy on behalf of myself and my co-authors.